# miR-199a Downregulation as a Driver of the NOX4/HIF-1α/VEGF-A Pathway in Thyroid and Orbital Adipose Tissues from Graves′ Patients

**DOI:** 10.3390/ijms23010153

**Published:** 2021-12-23

**Authors:** Julie Craps, Virginie Joris, Lelio Baldeschi, Chantal Daumerie, Alessandra Camboni, Antoine Buemi, Benoit Lengelé, Catherine Behets, Antonella Boschi, Michel Mourad, Marie-Christine Many, Chantal Dessy

**Affiliations:** 1Pole of Pharmacology and Therapeutics, Institute of Experimental and Clinical Research (IREC), Université Catholique de Louvain (UCLouvain), B-1200 Brussels, Belgium; julie.craps@uclouvain.be (J.C.); virginie.joris@uclouvain.be (V.J.); 2Pole of Morphology, Institute of Experimental and Clinical Research (IREC), UCLouvain, B-1200 Brussels, Belgium; benoit.lengele@uclouvain.be (B.L.); catherine.behets@uclouvain.be (C.B.); marie-christine.many@uclouvain.be (M.-C.M.); 3Department of Ophthalmology, Cliniques Universitaires Saint Luc, UCLouvain, B-1200 Brussels, Belgium; lelio.baldeschi@uclouvain.be (L.B.); antonella.boschi@uclouvain.be (A.B.); 4Departement of Endocrinology, Cliniques Universitaires Saint Luc, UCLouvain, B-1200 Brussels, Belgium; chantal.daumerie@uclouvain.be; 5Departement of Anatomopathology, Cliniques Universitaires Saint Luc, UCLouvain, B-1200 Brussels, Belgium; alessandra.camboni@uclouvain.be; 6Departement of Surgery and Abdominal Transplantation, Cliniques Universitaires Saint Luc, UCLouvain, B-1200 Brussels, Belgium; antoine.buemi@uclouvain.be (A.B.); michel.mourad@uclouvain.be (M.M.); 7Department of Plastic and Reconstructive Surgery, Cliniques Universitaires Saint-Luc, UCLouvain, B-1200 Brussels, Belgium

**Keywords:** Graves’ disease, Graves’ orbitopathy, microRNA-199a, oxidative stress, VEGF-A

## Abstract

Graves’ disease (GD) is an autoimmune thyroiditis often associated with Graves’ orbitopathy (GO). GD thyroid and GO orbital fat share high oxidative stress (OS) and hypervascularization. We investigated the metabolic pathways leading to OS and angiogenesis, aiming to further decipher the link between local and systemic GD manifestations. Plasma and thyroid samples were obtained from patients operated on for multinodular goiters (controls) or GD. Orbital fats were from GO or control patients. The NADPH-oxidase-4 (NOX4)/HIF-1α/VEGF-A signaling pathway was investigated by Western blotting and immunostaining. miR-199a family expression was evaluated following quantitative real-time PCR and/or in situ hybridization. In GD thyroids and GO orbital fats, NOX4 was upregulated and correlated with HIF-1α stabilization and VEGF-A overexpression. The biotin assay identified NOX4, HIF-1α and VEGF-A as direct targets of miR-199a-5p in cultured thyrocytes. Interestingly, GD thyroids, GD plasmas and GO orbital fats showed a downregulation of miR-199a-3p/-5p. Our results also highlighted an activation of STAT-3 signaling in GD thyroids and GO orbital fats, a transcription factor known to negatively regulate miR-199a expression. We identified NOX4/HIF-1α/VEGF-A as critical actors in GD and GO. STAT-3-dependent regulation of miR-199a is proposed as a common driver leading to these events in GD thyroids and GO orbital fats.

## 1. Introduction

Graves’ disease (GD) is an autoimmune thyroiditis that causes hyperthyroidism due to excess stimulation of thyrocytes by circulating thyrotropin (TSH) receptor-specific stimulatory autoantibodies (TRAb) [1]. While the TRAb antibodies provide an explanation for hyperthyroidism in GD, they do not fully account for structural changes seen in the progression of the disease, whose exact pathophysiology remains unknown [2]. The most common symptoms of GD are hyperthyroidism and highly vascularized goiter, but it is also known that GD is a multisystemic disease that can be associated with Graves’ orbitopathy (GO). In less severe cases, GO increases aperture of the palpebral fissure and mild exophthalmos, including red eyes, excess tearing and a feeling of sand in the eye. In more severe cases, remarkable increase in the eyelid aperture, remarkable exophthalmos, periorbital and conjunctival hyperaemia, and edema are observed. These symptoms mainly result from an increase in intraorbital pressure leading to secondary orbital and periorbital venous congestion [1,2]. A link between GD, GO and oxidative stress (OS) has already been suggested [3]. In tissues affected by the GD autoimmune process, OS and increased angiogenesis have been observed [3,4,5,6,7]. Antioxidant therapies, such as selenium supplementation, have been shown to slow down the progression of mild to moderately severe, non-active (clinical activity score < 2) GO of recent (<18 month) onset while combining conventional GD therapies with antioxidant supplements helps to normalize thyroid hormone levels faster [8].

Among the reactive oxygen species (ROS) responsible for OS, H_2_O_2_ is certainly a key player. H_2_O_2_ is crucial for thyroid hormone synthesis, and unbalanced production has been linked to thyroid autoimmune diseases [9,10,11,12]. Hence, Graves’ thyrocytes and orbital adipocytes showed increased 4-Hydroxynonenal protein expression (HNE), indicative of a rise in lipid peroxidation, and increased expression of catalase, an antioxidant enzyme, suggesting an enhanced H_2_O_2_ detoxification [4,12]. In the thyroid, H_2_O_2_ is mainly generated by DUOX2 [13,14] and to a lesser extent by NADPH oxidase (NOX) 4 [15]. In thyrocytes, NOX4 expression has been observed at the plasma membrane, endoplasmic reticulum, mitochondria and nucleus [16]. Interestingly, NOX4 is the only NADPH oxidase with constitutive ROS-generating activity, for which elevated mRNA levels directly reflect in vivo increased activity [17]. An upregulation of NOX4 has already been described in Hashimoto’s thyroiditis and in thyroid cancers with a particularly high expression in papillary thyroid cancers [15,18].

Previous studies proved that a variety of miRs are differentially expressed in GD [19,20]. Qin et al. highlighted a miR-target gene network in the pathogenesis of GD with 5 upregulated and 18 downregulated miRs in GD thyroid glands [21]. Some other miRs have also been proposed as biomarkers for cardiac aggravation or for therapeutic efficiency in GD patients [22,23]. We have recently highlighted that in endothelial cells, the miR-199a family, which comprises the mature forms miR-199a-3p and -5p, participates in a complex regulatory network targeting nitric oxide and ROS production and angiogenesis [24]. Interestingly, miR-199a-3p overexpression decreased lipid deposition and adipogenic gene expression in brown adipocytes [25], while miR-199a-5p upregulation reduced lipid accumulation in porcine preadipocytes [26]. Specifically in the thyroid, miR-199a-3p and -5p were proposed as tumor suppressors in papillary cancers (PTC) and follicular carcinoma, respectively [27,28]. Despite this established expression in thyroid tissues and their potential role in OS, angiogenesis and adipogenesis, nothing is known about the implication of the miR-199a family members in GD and GO.

We therefore investigated, in tissues obtained from patients, whether OS and angiogenesis observed in GD and GO could be associated with a dysregulation of the miR-199a family. Epigenetic modifications could indeed bring new perspectives in understanding the pathogenesis of GD and GO and be of potential clinical utility.

## 2. Results

GD thyroid histology was consistent with the activated state of the gland as compared to controls (see Appendix A). CD31 staining (Appendix A) highlighted numerous dilated capillaries, confirming the rich vascularization in GD thyroids. Among pathological changes in GO orbital fats (see Appendix A), expansion of the vascular network, as described by Görtz et al. [29], was also detected in our samples by CD31 immunohistochemistry (Appendix A). 4-Hydroxynonenal (HNE), a marker of lipid peroxidation and OS, was markedly detected in GD thyroids and GO orbital fat tissues in comparison to corresponding control tissues (Appendix A for GO versus respective controls in C and G), in agreement with previous observations made by Marique et al. [4,12]. According to Marique et al. [12], Caveolin (Cav)-1, which plays a key role in OS, was decreased in Hashimoto’s thyroiditis but not in GD. This was confirmed by our results showing an upregulation of the Cav-1 protein in GD by WB (Appendix A).

### 2.1. NOX4 Protein Expression Is Increased in GD and GO Tissues

Compared to control paranodular thyroid tissues of multinodular goiters (Figure 1A), the majority of the follicles from GD thyroids (Figure 1B) showed increased expression of NOX4 in thyrocyte’s cytoplasm. In adipocytes, cytoplasm staining for NOX4 was also pronounced in GO orbital fat (Figure 1D) versus the control (Figure 1C). Histological analyses were confirmed by WB, revealing a significant increase in NOX4 protein expression in GD thyroids and GO orbital fats compared to controls (Figure 1E,F). Of note, in both tissue types, endothelial cells only moderately expressed NOX4, regardless of the condition (Figure 1A1–D1).

### 2.2. HIF-1α and VEGF-A Protein Expression Is Increased in GD and GO Tissues

The transcription factor HIF-1α is known to be upregulated in conditions associated with hypoxia or increased OS [29]. In thyrocytes, among genes under the control of HIF-1α is the vascular endothelial growth factor isoform VEGF-A, which we already reported to be upregulated by early iodine deficiency-induced thyroid microvascular activation [30]. Immunodetection on GD thyroid samples showed a higher expression of HIF-1α than in control thyrocytes (Figure 2A,B). Orbital adipocytes similarly presented an upregulation of HIF-1α in GO compared to controls (Figure 2C,D). Endothelial cells likewise expressed HIF-1α in healthy and pathological tissues (Figure 2A–D,A1–D1). The significant increase in HIF-1α protein expression was further confirmed by WB in GD and GO tissues (Figure 2E,F).

GD thyrocytes and GO orbital adipocytes showed a higher expression of VEGF-A than controls (Figure 3A,B). In control orbital fat, VEGF-A was mainly expressed by endothelial cells, while in GO orbital fat tissues, this protein was also present in adipocytes (Figure 3C,D). However, the expression of VEGF-A in endothelial cells did not change with the clinical extension of the disease (controls versus GD/GO) (Figure 3A1–D1). WB of thyroids and orbital fats confirmed the significant upregulation of VEGF-A protein in GD and GO tissues (Figure 3E,F). Taken together, our results show an upregulation of the NOX4/HIF-1α/VEGF-A pathway in GD thyrocytes and GO orbital adipocytes.

### 2.3. NOX4, HIF-1α and VEGF-A mRNAs Are Direct Targets of miR-199a-5p in Thyrocytes

HIF-1α, NOX4 and VEGF-A have been proposed (based on in silico pairing, miRmap, Targetscan, pita, miranda) or have been identified as miR-199a direct targets in different tissues [31,32,33]. We then developed an assay to verify the existence of such regulation in GD. To do so, human primary thyrocytes were transfected with a biotinylated sequence of miR199-a-5p or scramble (bi-miR-199a-5p or with a bi-miR-scramble) and incubated for 48 h. Complexes of miR-199a-5p bound to targeted mRNAs were then collected using streptavidin coated beads. After RNA extraction, PCR amplification revealed that NOX4, HIF-1α and VEGF-A transcripts had indeed been precipitated in human thyrocytes transfected with bi-miR-199a-5p, similar to the input (Figure 4A). These three transcripts were not detected in thyrocytes transfected with bi-miR-scramble. Our results hence suggested that miR-199a-5p directly targets NOX4, HIF-1α and VEGF-A in human thyrocytes. By regulating NOX4, HIF-1α and VEGF-A, miR-199a could thus be considered a regulator of ROS production, OS and angiogenesis, as it has been proposed in other cell types [31,33,34,35,36] (Figure 4B).

### 2.4. miR-199a-3p and miR-199a-5p Are Downregulated in GD Thyroids

Expression of the miR-199a family was then investigated in control and GD thyroids. In situ hybridization showed intense (blue) specific cytoplasmic staining of miR-199-3p (Figure 5B) and miR-199-5p (Figure 5F) in control thyrocytes compared to the background colouringof the scramble probes (Figure 5A,E), showing that control thyrocytes highly expressed both miR-199-3p and -5p. In Graves’ thyroids (Figure 5D,H), the expression of miR-199-3p and -5p in the thyrocytes appears strongly reduced compared to controls (Figure 5D,H). miR199-3p and -5p were also detected in endothelial cells in both control and Graves’ thyroids, but their levels were not regulated by the pathology.

Our results showing the down-regulation of miR-199-3p and -5p were confirmed by qPCR in samples from Graves’ thyroids versus controls (Figure 5I,J).

### 2.5. A Higher Expression of VEGF-A and a Downregulation of miR-199a-3p and miR-199a-5p Are Observed in Human Primary Thyrocytes Treated with IL-4

To further characterize the molecular signaling beyond GD in thyrocytes, we used a model previously developed in our laboratory [12]. Briefly, human primary thyrocytes obtained from control paranodular tissues were treated with interleukin (IL)-4 to mimic GD. IL-4 treatment significantly increased VEGF-A protein expression in human primary thyrocytes (Appendix A). Additionally, although endothelial tube formation was limited, endothelial cell lining on Matrigel was enhanced upon exposure to a GD-conditioned medium (vs. control conditioned medium) (Appendix A). Importantly, in human primary thyrocytes treated with IL-4, a significant decrease of miR-199a-3p and -5p was also detected by qPCR, confirming the downregulation observed in native GD samples (Appendix A).

### 2.6. miR-199a-3p and miR-199a-5p Are Downregulated in GO Orbital Fat Tissues and in Plasma of GD Patients

In situ hybridization showed that miR-199-3p and 5p expression was decreased in adipocytes from GO patients compared to controls (Figure 6A–H). This again was confirmed by qPCR measurements in GO orbital fats compared to controls (Figure 6I,J). Interestingly, qPCR revealed no modification in miR-199a-3p and 5p expression between controls and GD subcutaneous adipose tissues (Figure 6K,L).

As it is known that some miR are able to circulate in the bloodstream, the expression of miR-199-3p and -5p was measured in the plasma of GD and control patients. A significant reduction of circulating miR-199a-3p and a decreasing trend of miR-199a-5p were observed in the plasma of GD patients (Appendix A).

### 2.7. STAT3 Activity Is Upregulated in GD Thyroids and GO Orbital Fats

Recent data showed that activation of transcription factor (STAT) 3 downregulates miR-199a-5p in hypoxic cardiomyocytes [32,37]. We therefore investigated STAT3 activation in GD thyroids and GO orbital adipose tissues through the quantification of the activation site at Tyrosine 705 (pTyr705 STAT3) (Figure 7A–F). STAT3 phosphorylation on Tyr705 increased significantly in GD thyroids (Figure 7A), while total STAT3 protein expression remained unchanged (Figure 7B). In GO orbital adipose tissue, an increasing trend was observed for STAT3 phosphorylation on Tyr705, while no modification was detected for total STAT3 protein expression (Figure 7D–F).

## 3. Discussion

Oxidative stress (OS) is a hallmark of GD and GO in link with a hypermetabolic state and high oxygen consumption. The inter-relationships between GD/GO and OS are complex and still incompletely understood [3], with OS being the cause and the consequence of the disease. The present work illustrates the role of NOX4 as a non-mitochondrial source of ROS in thyroids and orbital fats from GD/GO patients. Our work also uncovers the dysregulation of miR-199a as a common contributor to increased OS and angiogenesis in these tissues.

In thyrocytes, several defense mechanisms co-exist, allowing the cells to cope with the burden of ROS production, with the NOX/DUOX family composed of DUOX1, DUOX2, NOX2 and NOX4 being the major source of non-mitochondrial ROS [15,38,39]. An increase of catalase [12] and peroxiredoxin 5 [40] protein expression had been observed in GD together with a reduced activity of Superoxide dismutase and Glutathion reductase [41], but it is generally admitted that the OS level in GD/GO tissues depends more on an increase in ROS production than on a decrease in antioxidant defenses [11]. It has already been proven that NOX2 activation, inducing superoxide anion production, was increased in sera of hyperthyroid patients with the GD versus euthyroid state. This activation is linked to TRAb since no activation was detected in patients with non-autoimmune hyperthyroidism [42]. Our present results suggest that NOX4 also plays a key role in ROS production in GD/GO pathophysiology. Interestingly, in Hashimoto’s thyroiditis and in papillary thyroid cancer [43], thyroid NOX4 expression is also upregulated. In contrast to GD, in Hashimoto’s thyroiditis, Th1 cytokines appear to directly upregulate NOX4 and ROS production [18]. Previous results from our group also highlighted the upregulation of NOX2 in the cytoplasm of GO orbital adipocytes associated with eNOS activation and the loss of Cav-1, two key players in ROS generation in GO pathogenesis [44]. Our data here point to NOX4 as an additional player in GO orbital fat OS. NOX4 is the most studied NOX in adipocytes; its role in adipogenesis is however, controversial. Some data suggest that NOX4 acts as a negative regulator of adipocyte differentiation, while others proposed NOX4 as a driver of preadipocyte differentiation ([45,46] for a review, see De Vallance et al. [47]).

Hypervascularization has been described in GD and GO tissues [5,7] and it is already accepted that increased follicle activity is correlated with increased vascularization [48]. We demonstrate here that NOX4/HIF-1α/VEGF-A pathway is upregulated and this correlates with the local microvascular expansion in GD thyroids and GO orbital fats. NOX-generated ROS (including NOX4) are essential intermediates for HIF-1 activation associated with enhanced angiogenesis under non-hypoxic conditions [49,50]. Interestingly, a positive feedback loop (see Figure 4B) could reinforce the proangiogenic signaling cascade, as both NOX2 and NOX4 were shown to be targets of HIF-1α [34,51]. Similarly, in thyroid tumorigenesis, H_2_O_2_ produced by NOX4 increases HIF-1α activity, providing a feedback loop to modulate ROS production [52]. In addition, nitric oxide mediated VEGF-A upregulation via HIF-1α increase has been observed in thyrocytes in response to iodide deficiency [30,53]. Of note, in orbital fibroblasts derived from GO tissues, the link between HIF-1α/VEGF-A in response to hypoxia is made notably by Görtz et al. [29].

Several miR participate in GD pathogenesis [20]. The miR-199a family was not yet investigated. In situ hybridizationrevealed that both mature arms of miR-199a were largely expressed in control thyroids and orbital fats, with positive staining in thyrocytes, adipocytes and endothelial cells. Our data highlight a downregulation of miR-199a-3p/-5p in GD/GO tissues, especially in thyrocytes and orbital adipocytes. No changes were observed in endothelial cells from thyroids and orbital fats. This correlates with the absence of modification of miR-199a-3p/-5p levels in GD versus control subcutaneous veins (data not shown). The downregulation of miR-199a expression observed in GO orbital adipose tissue appears not only cell-specific but also location specific, as it was not present in GD subcutaneous adipose tissues. Although a decreasing trend of the miR-199a pattern was observed in plasma of GD patients (Appendix A), it is not known at this point from which tissue these circulating miRs originate and whether they participate in the control of their specific targets at distance from their production site(s). Two observations argue against this last hypothesis. The first is that we also observed a downregulation of miR-214 (data not shown), a miR expressed in cluster with miR-199a, suggesting local generation of the precursors (miR-199a2 and miR-214). The second counterargument arises from the upregulation of transcription factor STAT3 signaling, which was observed in the thyroid, as well as in orbital adipose tissue. Alteration in the phosphorylation status of STAT3 on its activation site (Tyr 705) correlates in these tissues with miR-199a repression, in line with the ability of STAT3 to negatively regulate miR-199a previously described in cardiomyocytes under hypoxic conditions [35,36,37,54]. This observation is particularly interesting in view of the recent data from Ko et al. who proposed STAT3 as a potential therapeutic target in GO [55]. They showed that STAT3 mRNA was increased in GO orbital adipose tissue compared to controls. Their in vitro results on GO orbital fibroblasts also suggests that STAT3 participates in the control of proinflammatory cytokine production, oxidative stress responses and adipogenesis. Another interesting link, although only described in lung cancer, is the positive feedback loop composed of miR-199a-5p/HIF1-α and STAT3 [36]. Of note, a post-transcriptional regulation could co-exist. Indeed, in the context of GO, anti-TSHR antibodies were shown to bind to TSHR on orbital fibroblasts, where they initiate signaling through Gαs and Gαq subunits [56]. The Gαq pathway mediates activation of phospholipase C (PLC) and leads to activation of phosphatidylinositol 3,4,5 triphosphate kinase (PI3K) and protein kinase B (AKT), elsewhere known to repressed miR-199a expression at the post-transcriptional level [57].

We had previously shown how miR-119a-3p and -5p participate in the regulation of angiogenesis and OS in the endothelium [24]. In the context of GD/GO, we did not observe any direct regulation in the endothelium per se. However, the dramatic upregulation of HIF-1α and VEGF-A could certainly account for the hypervascularization observed both in the thyroid and orbital adipose tissues from GD/GO patients. We demonstrated, in primary cultures of human thyrocytes, that NOX4/HIF-1α/VEGF-A mRNAs are direct targets of miR-199a-5p in thyrocytes, confirming the existence of epigenetic regulation of OS and angiogenesis by the miR-199a family in GD and GO. Whether other miR-199a targets participate in the pathophysiology remains to be investigated. One of them in the thyroid is probably Cav-1 (see Appendix A), previously reported as a target of miR-199a in lung fibroblasts and microvascular endothelial cells [58,59].

It is important to acknowledge that a downregulation of the miR-199a family, as well as an increase of NOX4, VEGF-A and HIF-1α expression, has also been described in PTC [60]. Epigenetic changes in the promoter region of miR-199a were indeed linked to severe repression of miR-199a-3p expression in thyroid tissues and were associated with PTC aggressiveness [27,60,61]. However, this common behavior regarding miR-199a regulation should not mask striking differences between GD and PTC phenotypes. Indeed, GD is characterized by cell hyperfunction and hyperthyroidism in response to TRAb, whereas PTC cells have lost their capacity for hormonal synthesis. Expression levels of Cav-1 are also of importance. Increased in GD thyrocytes, as shown by our results (Appendix A), they may participate in GD cell efficiency to synthesize T3 and T4, as Cav-1 is a key component of the thyroxisome, bringing together DUOX and TPO at the apical border of the thyrocyte [12]. Conversely, the decrease of Cav-1 in thyroid cancers [62] could lead to a disruption of the thyroxisome and to a loss of normal hormonal synthesis.

There are some limitations to our study analyzing thyroid and orbital fat tissues from patients who underwent surgery. The major limitation was the small number of tissue samples. Elective surgeries have often been postponed during the Covid pandemic, further increasing the difficulties in obtaining such samples. The limited number of patients precluded obtaining data on the influence of smoking or medication on the NOX4/HIF-1α/VEGF pathway or on miR-199a expression. Considering, on the one hand, the critical influence of smoking on the incidence, severity and response to treatment in Graves’ orbitopathy and, on the other hand, the very recent demonstration of a reduced miR-199a expression in human gingival fibroblasts exposed to cigarette smoke extract, the impact of smoking on NOX4/HIF-1α/VEGF and miR-199a warrant further investigations.

In conclusion, OS due to ROS overproduction (i.e., by NOX4) and upregulation of the NOX4/HIF-1α/VEGF pathway are primum movens features in GD–GO pathogenesis. By targeting NOX4, HIF1-α, and VEGF-A, miR-199a-5p could be considered a common actor of GD and GO.

## 4. Patients and Methods

### 4.1. Patients

For our study, we analyzed human thyroid tissues from Graves’ patients (*n* = 10) and paranodular thyroid tissues from patients operated on for multinodular goiter (*n* = 20). The thyroid tissues adjacent to the nodules present normal histological features and are thus considered control samples [12,18]. Orbital adipose tissues were obtained from GO patients who underwent orbital decompression surgery (*n* = 16) and were compared to control orbital adipose tissues (*n* = 4) removed because of subconjunctival orbital fat herniations or during blepharoplasties when orbital fat was to be removed because prolapsed through lax or ruptured septa. Subcutaneous adipose tissues and subcutaneous veins were also obtained from patients without (*n* = 27) or with (*n* = 11) GD. Analyses were not performed on all samples due to technical limits (mostly the limited size of the samples) and specific *n* are indicated in the results for each experiment. The surgical specimens and plasma were obtained after patients gave their informed consent and in agreement with our local ethics committee, which approved the study (2017/04OCT/466 and 2017/10OCT/473). Samples were kept at the Biolibrary of Cliniques Universitaires Saint-Luc (UCLouvain), referenced as BB190044, a member of the Biothèque Wallonie Bruxelles (BWB) and of biobanking and biomolecular resources research infrastructure in Belgium (BBMRI.be). One section of each human sample was conserved at −80 °C, while the remainder were fixed in 4% formaldehyde for histological sections.

Characteristics of patients with GD and GO included in this study are respectively reported in Table 1 and Table 2.

### 4.2. Human Thyroid Cell Cultures

Paranodular thyroid tissues from patients who underwent surgery for multinodular goiters (controls) were used to realize primary cultures of thyrocytes. Thyrocyte isolation was performed as previously described [30]. Cells were cultured in MEM medium containing 5% newborn calf serum, 1 mU/mL TSH, 100 U/mL penicillin-streptomycin and 2.5 µg/mL amphotericin in 6-well plates for 1 week. To mimic GD, thyrocytes were treated with interleukin-4 (IL-4) for 24 h. For the last 24 h prior to experiments, cultures were kept in low (0.5%) serum medium.

### 4.3. Transfection

Human primary thyrocytes were cultured in 6-well plates to about 70–80% confluence at the time of transfection. The negative control (scramble) or bi-miR-199a-5p mimic (Qiagen) were diluted at 40 nM in serum-free OptiMEM (Gibco) and mixed with Lipofectamine 3000 for 5 min to allow micelles formation, according to the manufacturer’s recommendation. Transfection mix was added to the cells (cultured in antibiotic-free medium) and incubated overnight. The medium containing the transfection reagent was then removed and replaced with fresh complete medium. The cells were used for the biotin pull-down assay 48 h after transfection.

### 4.4. Maxwell RNA Isolation

Thyroid and orbital fat tissues were crushed and homogenized in 200 μL of homogenization buffer containing 4 µL of thioglycerol. Then, 200 μL of lytic enhancer, 200 μL of lysis buffer and 30 μL of proteinase K were added to each sample and vortexed for 20 s before being incubated for 10 min at RT. A volume of 400 µL of plasma was mixed with 80 µL of Proteinase K and 230 µL of Lysis Buffer C. Then samples were vortexed for 5 s before being incubated for 15 min at 37 °C. Extractions were performed with the Maxwell^®^ RSC miRNA Tissue Kit (Promega, Cat # AS1460) or the Maxwell^®^ RSC miRNA plasma and serum kit (Promega, Cat # AS1680). Maxwell^®^ RSC Cartridges were prepared as described in the manufacturers’ protocol. After Maxwell purification, the RNA was stored at −80 °C.

### 4.5. microRNA Reverse Transcription and Quantitative Real-Time PCR (qRT-PCR)

Samples previously extracted with Maxwell technology were diluted to obtain an RNA solution of 20 ng/µL. The reaction mix was prepared according to miRCURY^®^ LNA^®^ RT manufacturer’s protocol (miRCURY LNA Universal RT microRNA PCR kits (Qiagen)). The reverse transcription step was carried out with Biometra^®^ UNO-thermoblock Thermal Cycler for 60 min at 42 °C and was followed by the step of an inactivating reaction for 5 min at 95 °C. The cDNA samples were diluted in RNAse-free water following manufacturer’s recommendations and 3 µL were engaged in qPCR. The reaction mix was prepared according to miRCURY^®^ LNA^®^ SYBR Green PCR Kit. The BioRad real-time PCR IQ5 was programmed to make 40 cycles after 2 min of initial heat activation at 95 °C. Each cycle consisted of 10 s of denaturation at 95 °C, 60 s of combined annealing and extension at 56 °C with a temperature rate of 1.6 °C/s. Experiments were performed in duplicate and levels were normalized based on U6 (tissue) or miR-Let-7i-5p (plasma) expression.

### 4.6. Western Blot Analysis

Human tissue samples, thyroid, orbital fat or isolated thyrocytes, were lysated in radioimmunoprecipitation (RIPA) buffer containing a protease inhibitor cocktail (Sigma) and PhoSTOP (Roche, Basel, Switzerland). A Bicinchoninic Acid (BCA) Protein Assay (Thermofisher Scientific, Waltham, MA, USA) was performed to determine the sample protein concentration according to the manufacturer’s protocol. Western blot (WB) analysis was performed as described previously [18,30]. Membranes were incubated at 4 °C overnight with the appropriate primary antibody (polyclonal Rabbit anti-NOX4 antibody: 14347-1-AP (Proteintech), polyclonal rabbit anti-HIF-1α antibody: NB 100,449 (Novus Bio), polyclonal rabbit anti-VEGF-A antibody: Abcam 46154polyclonal rabbit anti-phosphoTyr705-STAT3 antibody: CS9131 (Cell signaling), monoclonal mouse anti-STAT3 antibody: CS9139 (Cell signaling), or monoclonal mouse anti-Caveolin-1 antibody BD610407 (BD transduction) and followed by 1 h of incubation at RT with the peroxidase-conjugated secondary antibody. Revelation was performed by enhanced chemiluminescence (ECL) on CL-Xposure film (Pierce). Proteins of interest were normalized based on a loading control protein, β-actin or GAPDH.

### 4.7. Immunohistochemistry

As previously described [18], the sections were incubated in a citrate buffer (0.01 mol/L) and heated in a microwave oven according to the protocol: 750 W 3 min; 1 min break; 350 W 4 × 3 min 30 (1 min break between each). Sections were then placed in a cold-water bath for 15 min. Endogenous peroxidases were inactivated by incubation in 3% H_2_O_2_ for 20 min. The blocking step was performed by incubating the slides in 5% goat serum, Tris-buffered saline (TBS)–Tween 0.1%/5% bovine serum albumin (BSA) for 1 h at room temperature. Primary antibodies diluted in TBS-triton 0.1%/5% BSA (Polyclonal Rabbit anti-NOX4 antibody: 14347-1-AP (Proteintech), Polyclonal Rabbit anti-HIF-1α antibody: NB 100449 (Novus Bio), and Polyclonal rabbit anti-VEGF-A antibody: Abcam 46154) were incubated overnight at 4 °C. The anti-rabbit EnVision-HRP antibody (Dako, K4003) was added for 1 h, followed by a 5 min incubation with the diaminobenzidine (DAB) substrate kit peroxidase (Vector, SK-4100). Mayer hematoxylin counterstain (1–2 min) was finally performed.

### 4.8. In Situ Hybridization

Sections were incubated at 60 °C for 45 min and immersed in successive baths of toluol, ethanol and PBS for deparaffinization. Sections were then treated with proteinase K (15 μg/mL) for 30 min at 37 °C in a humid chamber. After a rinse step, hybridization mixtures containing miR-199a-3p or miR-199a-5p specific probes were applied on slides. Negative controls were realized for each type of tissue with LNA^TM^ probe Scramble-microRNA. Hybridization was carried out for 1 h in a humid chamber at 51 °C. Sections were washed in successive baths of different concentrations of saline sodium citrate buffer placed in a water bath at the hybridization temperature and in 2 baths at RT, according to the Qiagen manufacture’s protocol (miRCURY^®^ LNA^®^ miRNA ISH Optimization Kits FFPE).

A blocking solution was used for 15 min before applying the anti-DIG reagent (Anti-DIG-alkaline phosphatase antibody, 1/600) for 1 h in a humid chamber at RT. After washing slides in PBS-Tween, tissue samples were incubated with an alkaline phosphatase substrate for 2 h, and then the reaction was stopped with KTBT buffer (Tris-HCl (50 mM), NaCl (150 mM) and KCl (10 mM)). Sections were counterstained with Nuclear Fast Red^TM^ (Kernechtrot) for 1 min. After rinsing the slides in a running water bath for 10 min, a dehydration step was realized according to the manufacturers’ protocol. Results were analyzed the next day.

### 4.9. Biotin Pull-Down, Reverse Transcription, and Qualitative PCR

Human thyrocytes were collected in lysis buffer (20 mM Tris (pH7.5), 5 mM MgCl_2_, 100 mM KCl, 0.05% Tween 20, 50 U RNase OUT (Invitrogen) and protease inhibitor cocktails (Sigma)). M-270 streptavidin-coated magnetic beads (Invitrogen) were washed in lysis buffer and blocked for 2 h at 4 °C in lysis buffer containing 1% BSA and 1 mg/mL yeast tRNA. After two washes in lysis buffer, beads were gently mixed with sample lysates and incubated for 4 h at 4 °C on agitation. RNA linked to beads and input RNA were purified using the Trizol LS reagent protocol (Invitrogen) according to the manufacturer’s protocol and resuspended in 10 µL RNase-free water. 8 µL of extracted RNA were added to the cDNA synthesis reverse transcription reaction mixture containing 5× buffer (Promega), 0.5 mmol/L dNTP (Promega), 2 µM OligodT (Promega), 40 U RNase IN (Promega) and 200 U Moloney murine leukemia virus (M-MLV) reverse transcriptase (Promega). Reverse transcription was performed by incubation at 42 °C for 3 h, followed by an inactivating step of 5 min at 95 °C. Then, RNase-free water (20 µL) was added. Target mRNAs were detected by qualitative polymerase chain reaction (PCR) amplification on Biometra^®^ UNO-thermoblock Thermal Cycler. The reaction mixture contained 5 µL of cDNA, GoTaq green master mix (Promega) and 5 µM of each appropriate primer pair (VEGF: primer forward; 5′-GCAGATGTCCCGGCGAAGAGAAGA-3′ and primer reverse; 5′-CGGGGAGGGCAGAGCTGAGTGTTA-3′, HIF-1α: primer forward; 5′-CGCTTTCTCTGAGCATTCTGC-3′ and primer reverse; 5′-CCCTAACGTGTTATCTGTCGCT-3′ and NOX4: primer forward; 5′-CCAAGCAGGAGAACCAGGAGATT-3′ and primer reverse; 5′-AGGCCAGGAACAGTTGTGAAGAGA-3′). PCR amplification was performed in the following steps: 94 °C for 2 min, followed by 35 cycles of 94 °C for 30 s, 60 °C for 1 min, and 72 °C for 1 min, followed by a final step at 72 °C for 10 min. PCR transcripts were separated by agarose gel (1%) electrophoresis.

### 4.10. Statistical Analysis

Values were expressed as mean ± SEM. The Shapiro-Wilk normality test was performed, and statistically significant differences were determined using a Student t-test or a Mann-Whitney test for non-parametric data (GraphPad). A *p*-value of <0.05 was considered statistically significant.

## Figures and Tables

**Figure 1 ijms-23-00153-f001:**
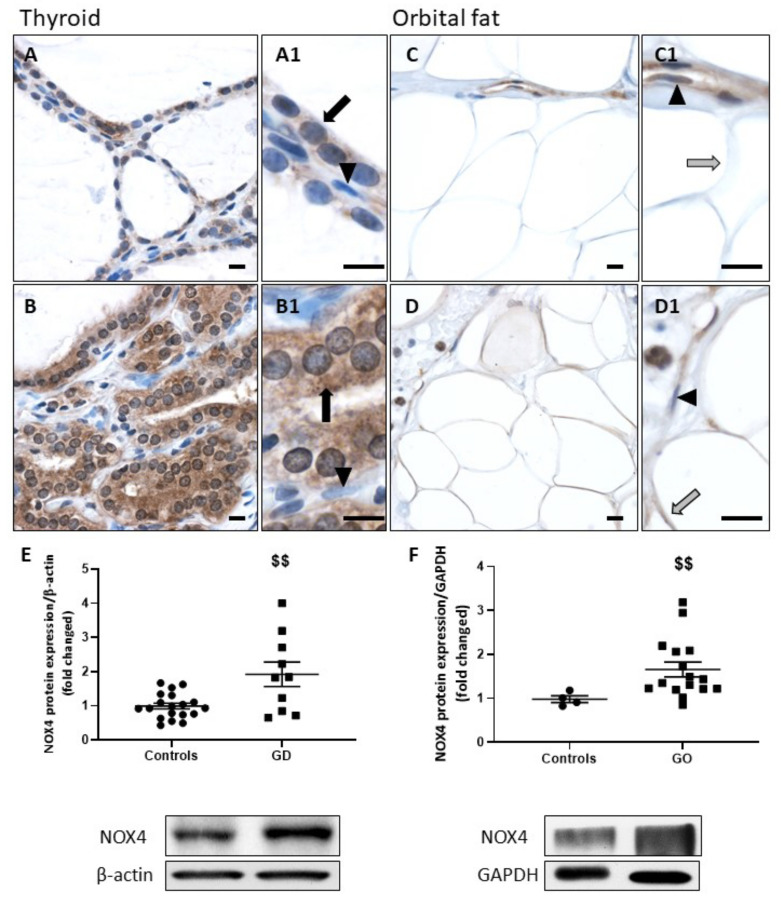
Increase of NOX4 in GD thyroids and GO orbital fats. In GD thyroids, NOX4 (**B**) was highly expressed in the thyrocyte cytoplasm. A weaker signal was detected in the cytoplasm of control thyroid cells (**A**). GO orbital adipocytes (**D**) also showed a higher expression of NOX4 than controls (**C**). A signal of similar intensity was detected in endothelial cells (arrowheads) in both conditions and tissues. (**A1**–**D1**) Thyrocytes are indicated by black arrows, adipocytes by gray arrows, and endothelial cells by arrowheads. Scale bars = 10 µm. Western blot analysis revealed that NOX4 was significantly increased in GD (**E**) or GO patients (**F**) versus controls. Data represent the mean ± SEM from 19 controls and 10 GD thyroids and from 4 controls and 16 GO orbital fats. $$ *p* < 0.01 compared to controls. Densitometric values were normalized against β-actin or GAPDH. Western blots shown are representative of both conditions.

**Figure 2 ijms-23-00153-f002:**
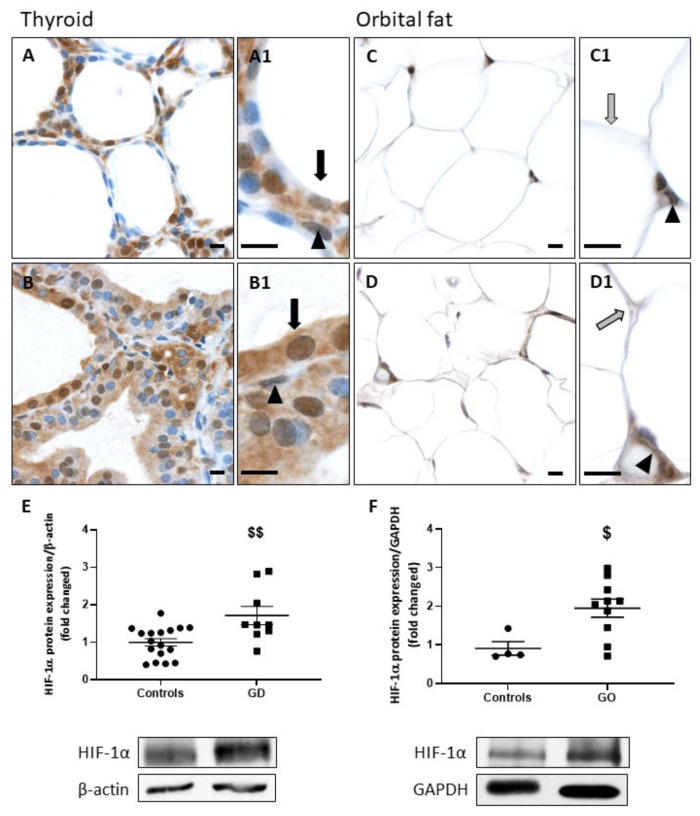
Increase of HIF-1α in GD thyroids and GO orbital fats. HIF-1α staining was higher in GD thyroids (**B**) than in controls (**A**). Similarly, GO orbital adipocytes presented a stronger expression of HIF-1α (**D**) compared to controls (**C**). Endothelial cells presented no difference in HIF-1α staining. (**A1**–**D1**) Thyrocytes are indicated by black arrows, adipocytes by gray arrows, and endothelial cells by arrowheads. Scale bars = 10 µm. Western blot analysis revealed that HIF-1α was significantly increased in GD (**E**) and GO patients (**F**). Data represent the mean ± SEM from 17 controls and 9 GD thyroids and from 4 controls and 10 GO orbital fats. $ *p* < 0.05 and $$ *p* < 0.01 compared to controls. Densitometric values were normalized against β-actin or GAPDH. Western blots shown are representative of both conditions.

**Figure 3 ijms-23-00153-f003:**
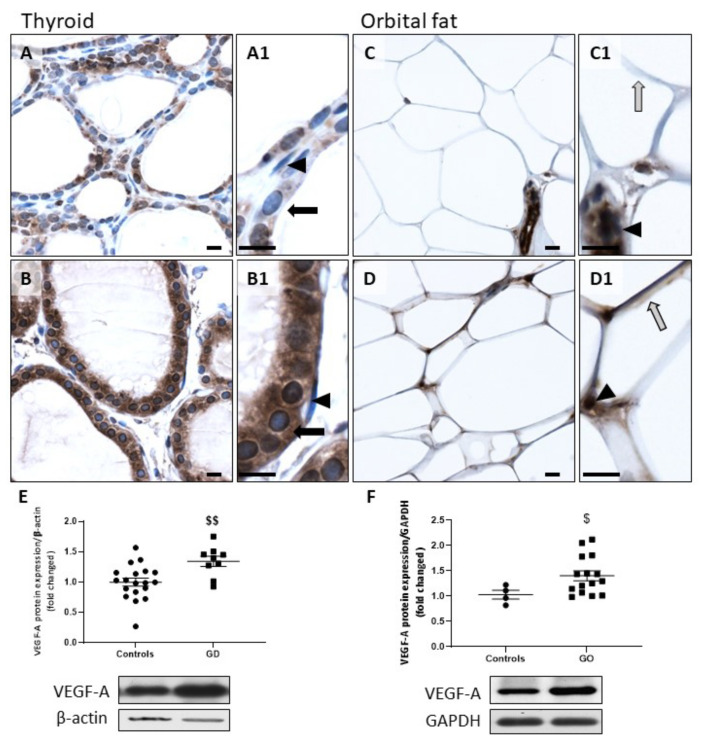
Increase of VEGF-A in GD thyroids and GO orbital fats. VEGF-A staining was predominant in GD thyrocytes (**B**) and GO orbital adipocytes (**D**) versus controls (**A**,**C**). Thyrocytes are indicated by black arrows, adipocytes by gray arrows, and endothelial cells by arrowheads (**A1**–**D1**). Scale bars = 10 µm. Western blot quantification of VEGF-A demonstrated an upregulation of this protein in GD (**E**) and GO (**F**) tissues. Data represent the mean ± SEM from 19 controls and 9 GD thyroids and from 4 controls and 15 GO orbital fats. $ *p* < 0.05 and $$ *p* < 0.01 compared to controls. Densitometric values were normalized against β-actin or GAPDH. Western blots shown are representative of both conditions.

**Figure 4 ijms-23-00153-f004:**
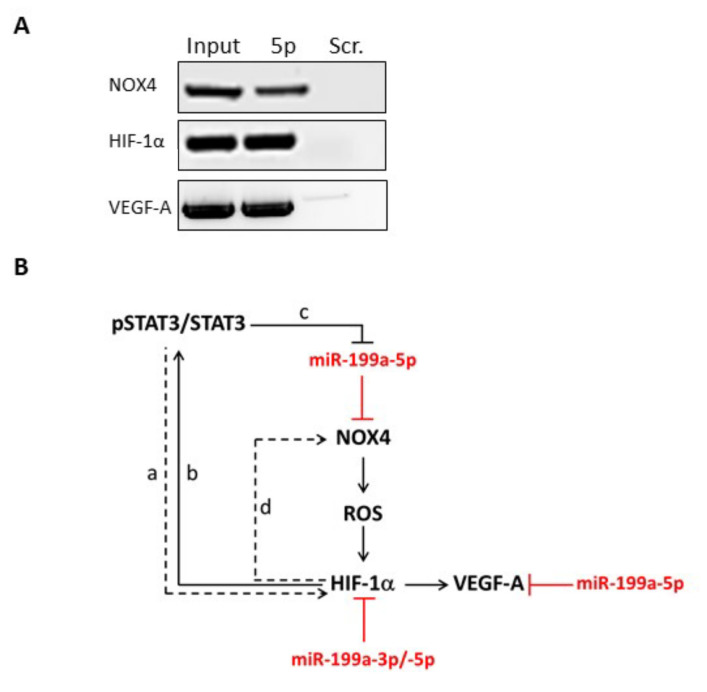
NOX4/HIF-1α/VEGF-A mRNAs are direct targets of miR-199a-5p in isolated thyrocytes. Human primary thyrocytes were transfected with either a negative control (scramble) or bi-miR-199a-5p mimic before realizing a biotin pull-down assay. NOX4/HIF-1α/VEGF-A mRNAs were detected in human thyrocytes transfected with the bi-miR-199a-5p mimic, while no expression was observed in thyrocytes treated with the scramble probe (**A**). By regulating NOX4/HIF-1α/VEGF-A pathway, miR-199a family could play a key role in ROS production, OS and angiogenesis in the context of GD and G0 (**B**) [4,10,12,29]. Black lines/arrows are for previously identified links (a—based on Jung [35]; b–c done by Yang [36]; d by Diebold [34]); red lines represent our working hypothesis.

**Figure 5 ijms-23-00153-f005:**
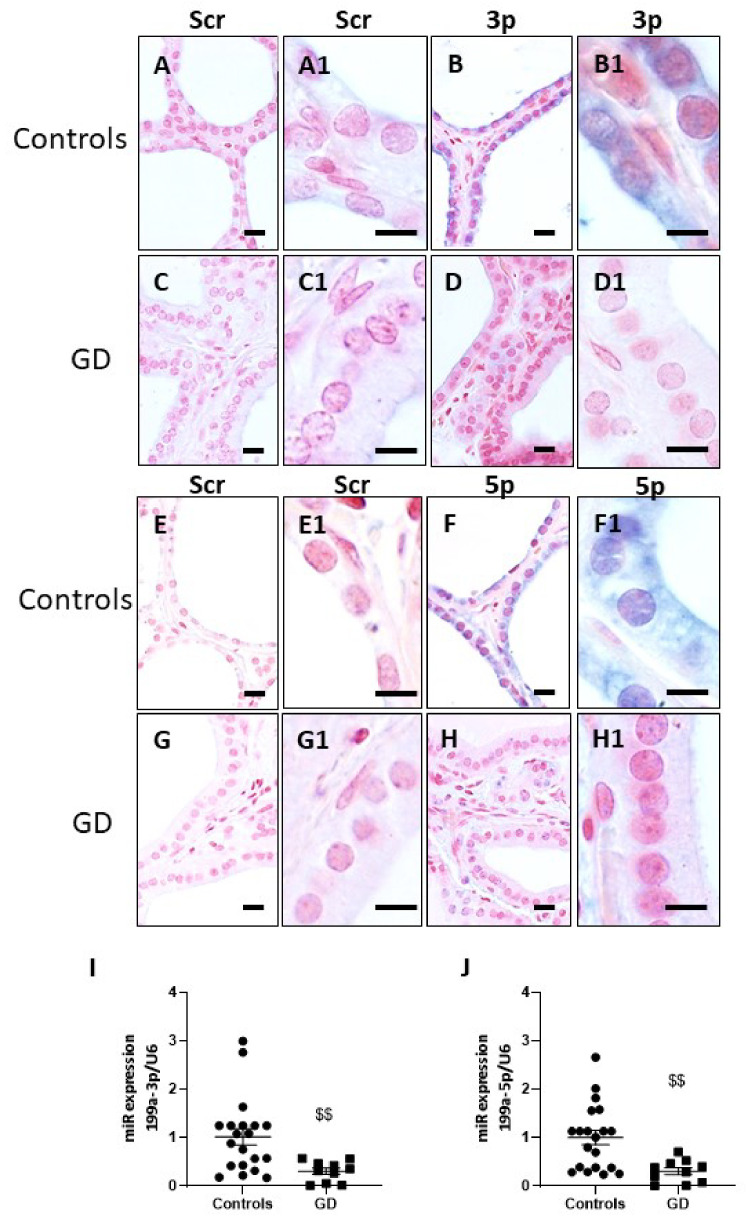
miR-199a family is downregulated in GD thyrocytes. In situ hybridization of LNA^TM^ probe Scramble-microRNA (**A**,**C**,**E**,**G**), of miR-199a-3p (**B**,**D**) and of miR-199a-5p expression (**F**,**H**) in control and GD thyroids. Note the weak staining in GD thyrocytes compared to controls (**A1**–**H1**). Scale bars = 10 µM. Measurement of miR-199a-3p and-5p by qRT-PCR showed a significant decrease of both miRs in GD thyroids (**I**,**J**). The data were normalized with the data of U6 snRNA. Data represent the mean ± SEM from 20 controls and 10 GD thyroids, $$ *p* < 0.01 compared to controls.

**Figure 6 ijms-23-00153-f006:**
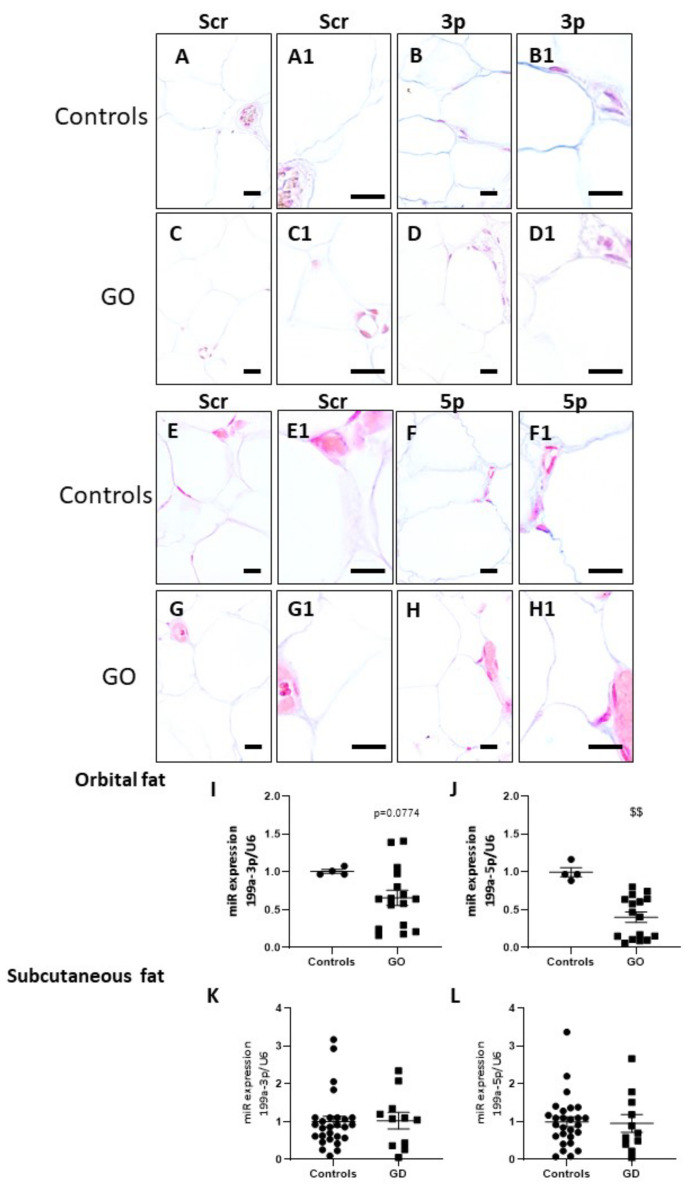
miR-199a family is downregulated in GO orbital adipocytes. In situ hybridization of LNA^TM^ probe Scramble-miR (**A**,**C**,**E**,**G**), of miR-199a-3p (**B**,**D**) and of miR-199a-5p expression (**F**,**H**) in controls and GO orbital fats. Note that control adipocytes expressed both miRs, whereas a very low signal was observed in GO orbital adipocytes (**A1**–**H1**). Scale bars = 10 µM. qRT-PCR confirmed a decreasing trend of miR-199a-3p and a significant reduction of miR-199a-5p in GO orbital fats (**I**,**J**). No modification of miR199a family profile expression was observed in GD subcutaneous adipose tissues compared to controls (**K**,**L**). The data were normalized with the data of U6 snRNA. Data represent the mean ± SEM for 4 controls and for 16 GO orbital fats and for 27 controls and 11 GD subcutaneous adipose tissues. $$ *p* < 0.01 compared to controls.

**Figure 7 ijms-23-00153-f007:**
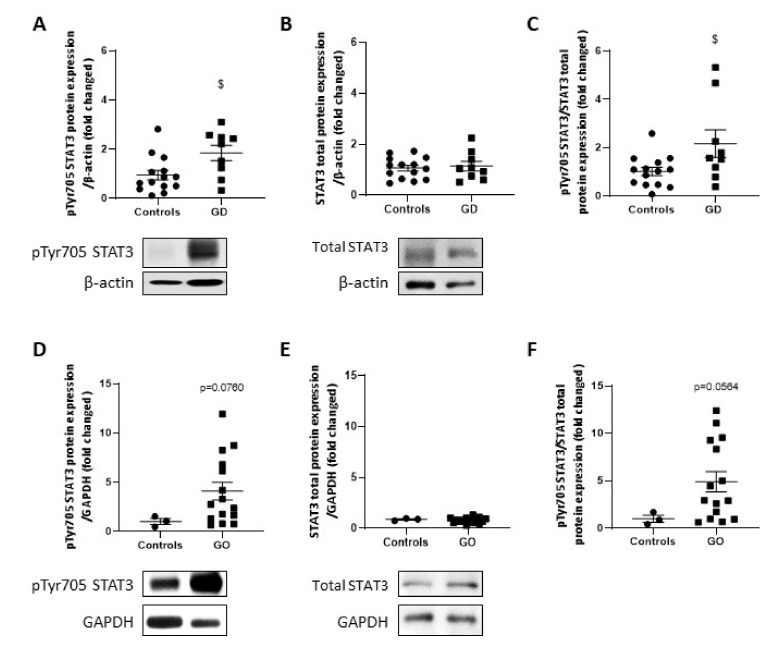
STAT3 is activated in GD thyroids and GO orbital fats. pTyr 705 STAT3 (**A**,**D**) and total STAT3 (**B**,**E**) were detected by Western blotting. Ratio of phosphorylated STAT3 to total STAT3 protein expression are represented in (**C**,**F**). A significant increase and an increasing trend of pTyr 705 STAT3 was observed in GD thyroids and GO orbital fats, respectively. Results are expressed as the mean ± SEM from 14 controls and 9 GD thyroids and 3 controls and from 15 GO orbital adipose tissues. $ *p* < 0.05 compared to controls. Densitometric values were normalized against β-actin for thyroid samples or GAPDH for adipose tissue. Western blots shown are representative of both conditions.

**Table 1 ijms-23-00153-t001:** Characteristics of patients with GD included in this study.

Number of GD patients		*n* = 10
Mean (range) age		34.4 (25–49) years
Sex	Female/Male	*n* = 8 / *n* = 2
Mean (range) GD duration		52.2 (3–156) months
Mean (range) TSH levels at biopsy time		1.13 (<0.01–2.33) mU/mL
Mean (range) TRAb levels at biopsy time		49.7 (1.7–151) U/mL

**Table 2 ijms-23-00153-t002:** Characteristics of patients with GO included in this study.

Number of GO patients		*n* = 16
Mean (range) age		52.0625 (24–75) years
Sex	Female/Male	*n* = 12/*n* = 4
Mean (range) GO duration		47 (4–240) months
Smoking habits	Smokers/Non-smokers	*n* = 10/*n*= 6
Mean (range) TSH levels at biopsy time		1.71 (0.005–3.15) mU/mL
Mean (range) TRAb levels at biopsy time		15.85 (0.3–33.5) U/L
Corticotherapy		*n* = 8
Mean (range) steroid therapy discontinuation time before surgery		5.5 (0–12) months

## Data Availability

Not applicable.

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
