# Peer review of "miR-199a Downregulation as a Driver of the NOX4/HIF-1α/VEGF-A Pathway in Thyroid and Orbital Adipose Tissues from Graves′ Patients"

_ijms, 2021, doi:10.3390/ijms23010153_

Round 1

Reviewer 1 Report

This is a detailed and convincing study.

Could you speculate on the interaction of the inflammatory pathways and the role of TSH-receptor antibody? Is there any known mechanism for TSH-RAb driven pathways to down-regulate mi199a?

You have pointed out that Hashimoto's thyroiditis shows activation of the NOX4 pathway, but presumably this pathway in HT is activated differently from Graves' disease?

Can you make any statement about differences in the pathway activation between smokers and non-smokers in your study, given the increased risk of GO and increased risk of GD hyperthyroidism and relapse in smokers?

Minor suggestion: in the title ' thyroid tissue' rather than "thyroids' is suggested to be better English expression.

Reviewer 2 Report

The manuscript analysed metabolic pathways leading to oxidative stress and angiogenesis aiming to elucidate the link between local and systematic Graves’ disease manifestation. Materials for the performed analyses were taken from a sufficient number of patients. A number of molecular methods were used to confirm the results. The obtained results are very interesting and well presented. I find one small mistake in reference 18 – number (8) is unnecessary. The manuscript can be accepted in its present form.

Reviewer 3 Report

The paper describes a very comprehensive study aimed to show the role of NADPH-oxidase-4 (NOX4)/HIF-1α/VEGF-A signaling pathway as well as miR-199a downregulation in the pathophysiology of Graves’ disease and Graves’ orbitopathy. The manuscript is well structured and well written. I especially appreciate the way the authors presented a rather complex set of results. The study is well designed to show/prove the concept. But there is a small fly in the ointment. I missed a paragraph on the limitations of the study. Obviously, the number of analyzed tissue samples was small, which can be explained by difficulties in obtaining such samples. But there is another issue omitted: What was the possible influence of the medical treatment used? The only data presented in this respect is that a half of patients with orbitopathy received steroids (for how long, in what doses?). And there is no consideration for antithyroid drugs. Don’t they influence ROS generation? Or even the NOX4/HIF-1α/VEGF-A signaling pathway? I would like to see authors’ comment on that.

Minor remarks: The paper was carefully prepared but I have noticed several typing errors: 

Line 129 – it should be “blot” not “bot”

Line 297 consider: the most studied NOX in adipocytes 

Line 345 it should be “could certainly account”

Line 468 – endly? Is there such a word?
